# LynA regulates an inflammation-sensitive signaling checkpoint in macrophages

**Tanya S Freedman[1,2]\*, Ying X Tan[1†], Katarzyna M Skrzypczynska[1],
Boryana N Manz[1], Frances V Sjaastad[2], Helen S Goodridge[3], Clifford A Lowell[4],
Arthur Weiss[1,5]\***

[1]Russell/Engleman Rheumatology Research Center, Division of Rheumatology, Department of Medicine, University of California, San Francisco, San Francisco, United States; [2]Department of Pharmacology, Center for Immunology, University of Minnesota, Minneapolis, United States; [3]Regenerative Medicine Institute and Research Division of Immunology, Cedars-Sinai Medical Center, Los Angeles, United States; [4]Department of Laboratory Medicine, University of California, San Francisco, San Francisco, United States; [5]Howard Hughes Medical Institute, University of California, San Francisco, Chevy Chase, United States

**Abstract** Clustering of receptors associated with immunoreceptor tyrosine-based activation motifs (ITAMs) initiates the macrophage antimicrobial response. ITAM receptors engage Src-family tyrosine kinases (SFKs) to initiate phagocytosis and macrophage activation. Macrophages also encounter nonpathogenic molecules that cluster receptors weakly and must tune their sensitivity to avoid inappropriate responses. To investigate this response threshold, we compared signaling in the presence and absence of receptor clustering using a small-molecule inhibitor of Csk, which increased SFK activation and produced robust membrane-proximal signaling. Surprisingly, receptor-independent SFK activation led to a downstream signaling blockade associated with rapid degradation of the SFK LynA. Inflammatory priming of macrophages upregulated LynA and promoted receptor-independent signaling. In contrast, clustering the hemi-ITAM receptor Dectin-1 induced signaling that did not require LynA or inflammatory priming. Together, the basal-state signaling checkpoint regulated by LynA expression and degradation and the signaling reorganization initiated by receptor clustering allow cells to discriminate optimally between pathogens and nonpathogens.

**\*For correspondence:**
tfreedma@umn.edu (TSF); art.weiss@ucsf.edu (AW)

**Present address:** †Institute of Molecular and Cell Biology, Agency for Science, Technology and Research, Singapore, Singapore

**Competing interests:** The authors declare that no competing interests exist.

## Introduction

Macrophages are myeloid-derived hematopoietic cells that guard against infection and orchestrate a broad immune response to pathogens. Bridging the innate and adaptive immune systems, they survey tissues for signs of pathogen invasion and present antigen to T cells. When activated, they engulf and destroy foreign organisms and secrete cytokines (*Mosser and Edwards, 2008*; *Murray and Wynn, 2011*; *Wynn et al., 2013*). While macrophage activity promotes an effective immune response to tissue distress, inappropriate activation of macrophages can cause tissue injury. Accumulation of activated macrophages is linked to diseases such as rheumatoid arthritis, type 1 diabetes, and obesity (*Murray and Wynn, 2011*; *Wynn et al., 2013*).

Macrophage stimulation through immunoreceptor tyrosine-based activation motif (ITAM)- and hemi-ITAM-coupled receptors initiates pathogen destruction via phagocytosis and the production of reactive oxygen species, which require activated macrophages to be physically close to the pathogen. To prevent nonproductive activation and tissue damage, macrophages must differentiate between

**eLife digest** Macrophages are white blood cells that protect the body from infection by bacteria and other microbes. Proteins and sugars on the microbe bind to receptor proteins on the surface of the macrophage, which triggers the macrophage to engulf the cell and destroy it. Macrophages also release molecules that are toxic to the microbe and activate other immune responses in the body. It is vital that macrophages can tell the difference between normal body cells and microbes because if macrophages are activated at the wrong time, they can damage tissues and cause inflammatory diseases.

When the receptor proteins are activated by contact with a microbe, they interact with a family of proteins called SFKs, which in turn stimulate communication systems called signaling pathways inside the macrophages that activate their immune responses. However, these responses are only triggered if many receptors are activated and clustered together. Thus, macrophages are able to identify whether a cell is a normal part of the body or is a foreign invader by sensing the number of receptors that cluster together as they bind to the cell. However, it is not clear if it is the clustering itself that signals a genuine encounter with a microbe, or whether this information comes from the strength of the response by the SFK proteins.

Here, Freedman et al. addressed this question by developing a new experimental system that allows SFKs to be directly activated in macrophages in the absence of receptor clusters. The experiments used macrophages obtained from genetically engineered mice and found that the direct activation of SFK proteins alone does not fully activate signaling pathways in the macrophages. A signaling blockade occurs due to the rapid destruction of an SFK protein called LynA. When the macrophages were exposed to molecules that are signals of inflammation in the body, they produced more LynA. This allowed these macrophages to be activated even without the formation of receptor clusters by interactions with pathogens.

Freedman et al.'s findings reveal that LynA acts as a checkpoint that primes macrophages to respond more aggressively when the body is under attack. Future work will be aimed at understanding how LynA is destroyed and how the detection of real pathogens overcomes the checkpoint.

receptor ligands displayed during a pathogen encounter (e.g., a β-glucan-presenting fungal cell wall or an IgG-opsonized cell) vs. environmental noise (e.g., glycans shed from fungi or other cell debris). One mechanism by which cells make this distinction is by using receptor clustering to sense the size of a potential stimulus. Large-diameter stimuli that are more likely to be associated with pathogenic particles nucleate large-scale receptor clusters, forming a signaling complex called a phagocytic synapse. In contrast, small-diameter stimuli induce only small-scale or transient receptor clusters (*Goodridge et al., 2011*). Hereafter, we will refer to ligands that cluster ITAM/hemi-ITAM receptors efficiently as 'strong stimuli' and ligands that cluster receptors poorly as 'weak stimuli'. In mast cells, a related myeloid subtype, treatment with a high dose of weak stimulus can produce robust receptor phosphorylation without stable receptor clustering. Interestingly, the resulting membrane-proximal signaling patterns, as well as the associated immunopathologies, are qualitatively different than those produced in response to a strong stimulus (*Carroll-Portillo et al., 2010*; *Suzuki et al., 2014*), which indicates that myeloid cells can distinguish between strong and weak stimuli beyond simply sensing the degree of receptor phosphorylation.

Receptor engagement initiates interactions with an active pool of Src-family tyrosine kinases (SFKs), which phosphorylate intracellular ITAMs within the receptor, the tyrosine kinase Syk, cytoskeleton-associated proteins, and other membrane-proximal signaling proteins (*Lowell, 2010*; *Goodridge et al., 2011*). This membrane-proximal signaling initiates downstream signaling, including the phospholipase Cγ-mitogen-activated protein kinase (PLCγ-MAPK) and phosphoinositide 3-kinase-protein kinase B/Akt (PI3K-Akt) pathways. The signaling cascades initiated by the SFKs in response to strong stimuli then enact a transcriptional program of macrophage activation leading to microbicidal activity (*Lowell, 2010*; *Byeon et al., 2012*).

To some extent, macrophages can modulate their responsiveness to activating stimuli according to their environment. For example, inflammatory cytokines secreted by other immune cells trigger a process termed inflammatory priming, a transcriptional program that increases macrophage

reactivity. Priming is known to upregulate the expression of proteins involved in signal detection, pathogen killing, and stimulation of other immune cells (*Platanias, 2005*). However, it is not clear how inflammation might modulate macrophage responses to weak stimuli by changing the extent of receptor clustering necessary for downstream signaling.

It is also unclear whether strong macrophage stimuli are defined by their ability to robustly activate the SFKs or whether receptor clustering itself relays qualitative information that indicates a genuine pathogen encounter. Clustered receptors are immobilized within the cell membrane, participate in costimulatory interactions, and organize the signaling complex spatially, integrating SFK-dependent and SFK-independent steps into a productive signaling response (*Sohn et al., 2008*; *Bethani et al., 2010*; *Dustin and Groves, 2012*). Until now, it has not been possible to study the contributions of SFK activity to macrophage activation independently of the integrated effects of receptor clustering.

To investigate the role of SFK activation in macrophage signaling independently of ligand-induced receptor clustering, we developed an experimental system to activate the SFKs directly by inhibiting their negative regulator, the tyrosine kinase Csk. Csk phosphorylates the inhibitory-tail tyrosine of the SFKs, promoting an autoinhibited state in which the SH2 domain of the SFK interacts with the phosphorylated inhibitory tail and constrains the kinase domain in an inactive conformation (*Brown and Cooper, 1996*; *Okada, 2012*). The phosphatases CD45 and CD148 dephosphorylate the inhibitory-tail tyrosine of the SFK, releasing this autoinhibition and enabling subsequent activation-loop autophosphorylation and full activation (*Chow and Veillette, 1995*; *Zhu et al., 2011*). The continuous competition of Csk and CD45/CD148 sets the steady state of SFK inhibitory-tail phosphorylation in the cell. Inhibiting Csk activity disrupts this steady state, allowing CD45 and CD148 initiate bulk SFK activation unopposed (*Schoenborn et al., 2011*). We previously described a Csk variant (analog-sensitive Csk, 'Csk$^{AS}$') that is sensitized to inhibition by a 3-iodobenzyl analog of the kinase inhibitor PP1 (3-IB-PP1). The bulkiness of 3-IB-PP1 prevents its interaction with WT Csk and other WT kinases (*Okuzumi et al., 2010*; *Schoenborn et al., 2011*; *Tan et al., 2014*). We have also described a transgenic mouse that expresses Csk$^{AS}$ but not WT Csk and have studied the effects of Csk$^{AS}$ inhibition in T cells (*Schoenborn et al., 2011*; *Tan et al., 2014*). Macrophages express a distinct subset of Src family members (Lyn, Hck, and Fgr) (*Lowell, 2004*), and, unlike naive T cells, macrophages express the ITAM-proximal kinase Syk (*Wang et al., 2010*) and SFK-dependent negative-regulatory phosphatases associated with immunoreceptor tyrosine-based inhibitory motifs (ITIMs) (*Veillette et al., 2002*; *Lowell, 2004*). This distinct membrane-proximal signaling environment likely modifies the signaling program initiated by macrophage receptors in service of cell-specific functions.

We derived macrophages from the bone marrow of *Csk$^{AS}$* mice and compared SFK signaling without receptor clustering (by inhibiting Csk$^{AS}$ with 3-IB-PP1) to SFK signaling induced by receptor clustering (by ligating the hemi-ITAM receptor Dectin-1 with depleted zymosan). Receptor-independent SFK activation by 3-IB-PP1 induced robust membrane-proximal signaling but no downstream signaling through the MAPKs or Akt. We determined that this signaling blockade was caused by rapid degradation of the SFK LynA, which resulted in a loss of function that could not be compensated for by the other SFKs. We were able to rescue downstream signaling by priming the macrophages, which led to the upregulation of LynA. Receptor clustering enabled the participation of the other SFKs in the activation of downstream MAPK, Akt, and calcium signaling independently of LynA. From the data presented in this article, we propose a model to explain how macrophages are prevented from responding to weak stimuli, how inflammation increases macrophage sensitivity to weak stimuli, and how receptor clustering rewires SFK signaling to enable macrophage activation.

## Results

### SFK activation in the absence of receptor clustering fails to induce downstream signaling

#### Inhibiting Csk in macrophages leads to rapid SFK activation

We generated bone marrow-derived macrophages (BMDMs) from *Csk$^{AS}$* mice and verified that they express normal levels of myeloid and macrophage surface markers (*Figure 1*). Within three seconds of adding 3-IB-PP1 to Csk$^{AS}$ BMDMs, we observed a 60–80% loss of phosphorylation of the SFK inhibitory-tail tyrosine and a 100–400% increase in activation-loop tyrosine phosphorylation (*Figure 2*,

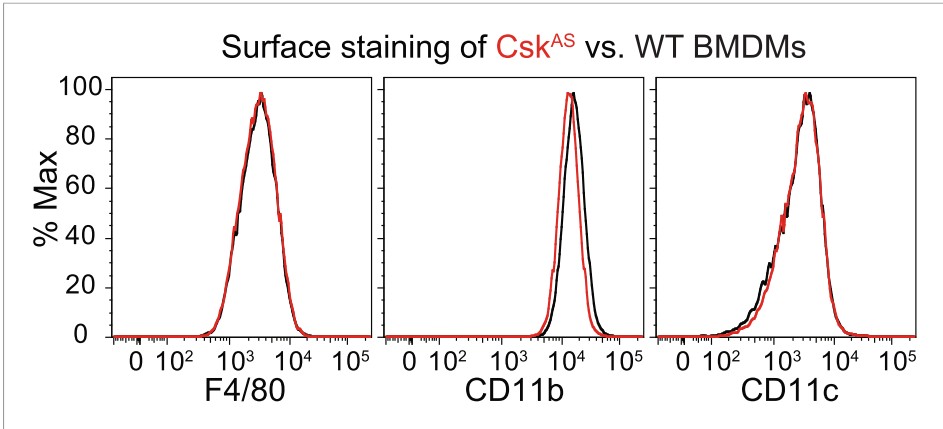

**Figure 1**. Surface-marker expression of Csk[AS] BMDMs. Expression of the surface markers F4/80, CD11b, and CD11c in bone marrow-derived macrophages (BMDMs) from Csk[AS] mice was assessed by flow cytometry. Data in this figure and those that follow are representative of three or more independent experiments.

left lanes). Activated SFKs continued to accumulate, reaching a maximum fivefold to eightfold above basal within 90 s. As expected from the low affinity of 3-IB-PP1 for WT Csk (*Tan et al., 2014*), 3-IB-PP1 treatment had no effect on SFK phosphorylation in WT BMDMs (*Figure 2*, right lanes).

## Activated SFKs initiate robust membrane-proximal signaling but no downstream signaling

We next examined signaling downstream of the SFKs in the presence and absence of receptor clustering. To investigate signaling in response to receptor clustering, we treated macrophages with zymosan, a particulate β-glucan derived from yeast cell walls that binds the Dectin-1 hemi-ITAM receptor (*Underhill, 2003*; *Goodridge et al., 2011*). The preparations of zymosan used for our experiments were depleted of TLR2 agonists, and this depleted zymosan is hereafter referred to as zymosan[dep] (*Figure 3*, *Figure 3—figure supplement 1*). To initiate and synchronize signaling, zymosan[dep] particles were settled onto adherent macrophages by pulse spinning. As expected, treatment with zymosan[dep] induced phosphorylation of the MAPK Erk as well as phosphorylation of Akt (*Figure 3*). Abrogation of downstream signaling in the presence of the Syk inhibitor BAY 61-3606 (*Figure 3A*) and the SFK inhibitor PP2 (*Figure 3B*) confirmed the dependence of zymosan[dep] signaling on SFK and Syk activation, especially within the first 5 min of signaling before Syk begins to be activated independently of the SFKs (*Takata et al., 1994*; *Fitzer-Attas et al., 2000*).

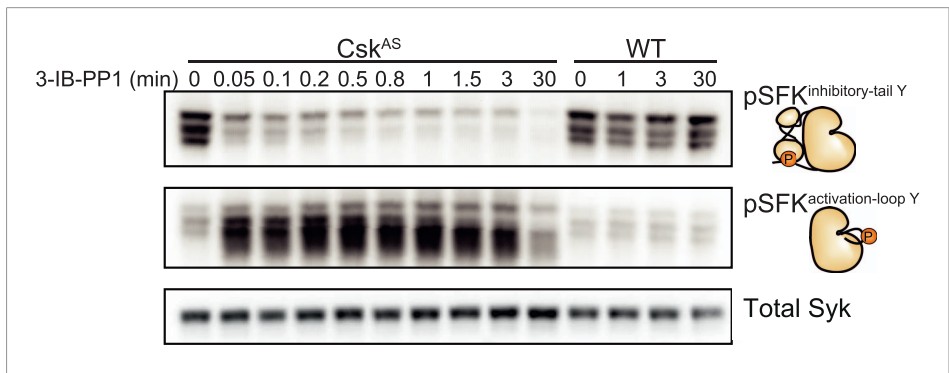

**Figure 2**. Csk inhibition leads to rapid activation of the SFKs. Adherent BMDMs generated from *Csk[AS]* or *WT* mice were treated with 10 μM 3-IB-PP1. The resulting lysates were separated by SDS-PAGE and subjected to immunoblotting with antibodies specific to the inactive and active forms of the Src-family tyrosine kinases (SFKs) (pLyn[Y507] and pSFK[Y416], respectively). An immunoblot of total Syk protein shows the total protein content in each lane.

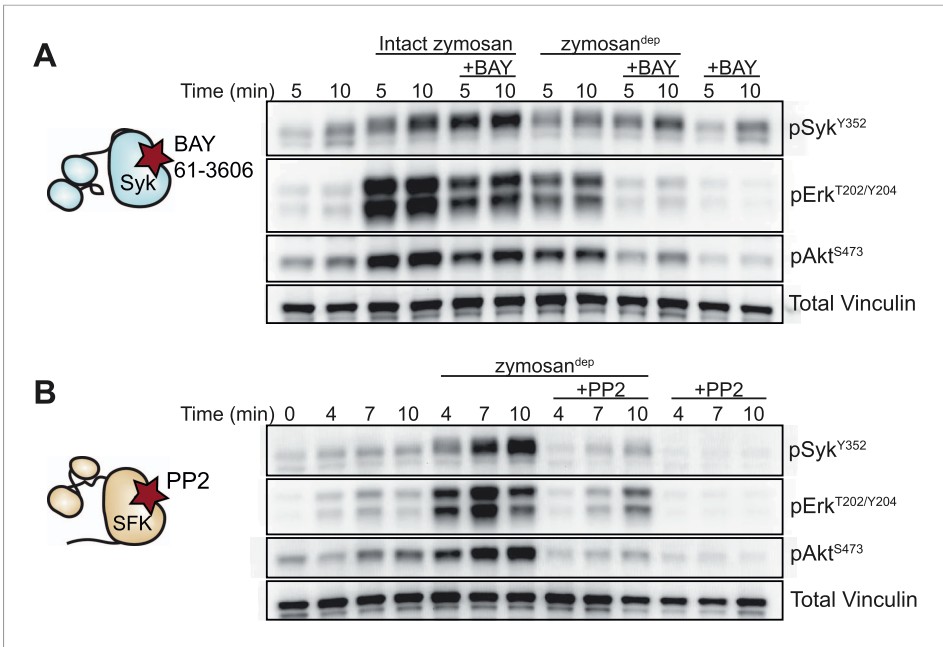

**Figure 3**. Depleted zymosan signals through the Src-family and Syk kinases. (**A**) BMDMs were pulse-spun with intact zymosan or zymosan[dep] (10 particles per cell) in the presence and absence of the Syk inhibitor BAY 61-3606 (1 µM). Signal transduction was assessed by immunoblotting with antibodies specific to activating phosphorylation sites of Syk, Erk, and Akt. Vinculin immunoblots are shown as loading controls. (**B**) The effect of the SFK inhibitor PP2 (20 µM) on zymosan[dep] stimulation was also assessed. See *Figure 3—figure supplement 1* for a model of signaling induced by intact and depleted zymosan.

The following figure supplement is available for figure 3:

**Figure supplement 1**. Signaling through intact and depleted zymosan has different requirements for Syk activation.

In contrast to zymosan[dep], SFK activation by 3-IB-PP1 failed to induce significant levels of Erk phosphorylation (*Figure 4A*, arrow). Small changes in Erk phosphorylation were occasionally seen in response to 3-IB-PP1 treatment. However, within each experiment, any minimal increase in Erk phosphorylation with 3-IB-PP1-treatment was equivalent to the effect of pulse-spinning the cells without any additional stimulation (*Figure 4B*).

Interestingly, upstream activators, including the SFKs, Syk, and PLCγ2, were phosphorylated robustly in response to 3-IB-PP1 treatment (*Figure 4A*). In fact, SFK activation by this maximally effective dose of 3-IB-PP1 (*Tan et al., 2014*) produced much more robust membrane-proximal signaling than did zymosan[dep], where an increase in SFK activation-loop phosphorylation was not even detectable. It is possible that our failure to detect SFK activation-loop phosphorylation in response to zymosan[dep] was due to only a small, local pool of SFKs being activated at the site of receptor clustering, as suggested by previous imaging (*Goodridge et al., 2011*). Alternatively, perhaps an increase in SFK activation is not the initiating event of receptor-mediated signaling. Rather, it is the interaction of clustered receptors with a preexisting pool of active SFKs, whose function is somehow blocked outside of large-scale receptor clusters. This model would be consistent with recent studies in T cells in which a basally active pool of SFKs is the major driver of TCR signaling (*Nika et al., 2010*; *Manz et al., 2015*).

The bulky structure of 3-IB-PP1 should increase its specificity for Csk[AS] over WT kinases, and we have shown experimentally that 3-IB-PP1 is 30-fold more selective for Csk[AS] than for WT Csk (*Tan et al., 2014*). However, all ATP-mimetic kinase inhibitors, even the bulkier analogs, have the potential for nonspecific interactions with kinase active sites (*Bain et al., 2007*). To minimize the possibility that an off-target effect of 3-IB-PP1 blocks Erk phosphorylation by a Csk[AS]-independent mechanism, we treated WT BMDMs with a combination of zymosan[dep] and increasing concentrations of 3-IB-PP1 and

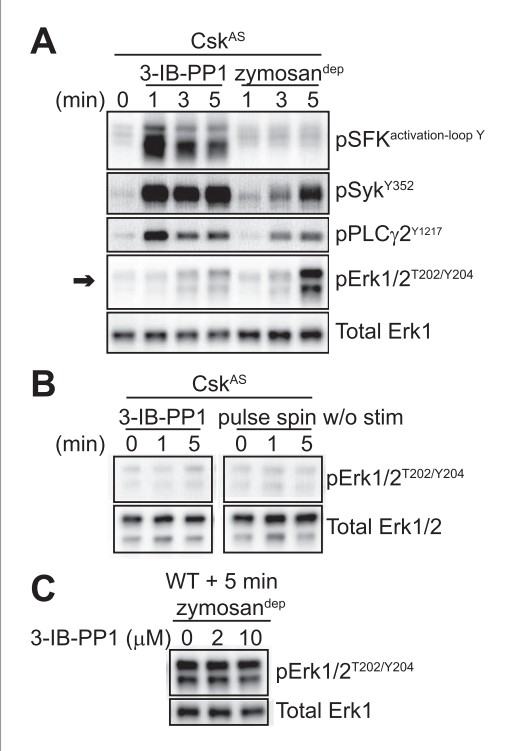

**Figure 4**. SFK activation in the absence of receptor clustering fails to stimulate downstream signaling in BMDMs. (**A**) Csk[AS] BMDMs were pulse-spun with 3-IB-PP1 (10 μM) or with zymosan[dep] (10 particles per cell). Signal transduction was assessed by immunoblotting with antibodies specific to activating phosphorylation sites of the SFKs, Syk, PLCγ2, and Erk (arrow). Total Erk1 is shown as a loading control. (**B**) Csk[AS] BMDMs were pulse-spun in the presence and absence of 3-IB-PP1, and Erk phosphorylation was assessed by immunoblot. Total Erk1/2 is shown as a loading control. (**C**) WT BMDMs were treated with zymosan[dep] for 5 min with varying concentrations of 3-IB-PP1 and analyzed by immunoblot. See *Figure 4—figure supplement 1* for additional tests of 3-IB-PP1 specificity for Csk[AS], *Figure 4—figure supplement 2* for evidence of that 3-IB-PP1 cotreatment suppresses signaling through the CSF-1 receptor, and *Figure 4—figure supplement 3* for evidence that actin-remodeling agents cannot restore 3-IB-PP1-induced downstream signaling.

The following figure supplements are available for figure 4:

**Figure supplement 1**. Further tests of 3-IB-PP1 specificity for Csk[AS].

**Figure supplement 2**. Inhibitory effect of 3-IB-PP1 on M-CSF-induced signaling in Csk[AS] cells.

**Figure supplement 3**. Actin-remodeling agents do not restore signaling downstream of 3-IB-PP1 treatment.

found no suppression of Erk phosphorylation (*Figure 4C*). Signaling in response to the growth factor M-CSF, which initiates Erk and Akt signaling through the CSF-1 receptor tyrosine kinase pathway (*Pixley and Stanley, 2004*), was similarly unaffected by 3-IB-PP1 in WT BMDMs (*Figure 4—figure supplement 1A*). We also tested 3-IB-PP1 specificity by treating WT and Csk[AS] BMDMs with 3-IB-PP1 and assessing global tyrosine phosphorylation by immunoblotting (*Figure 4—figure supplement 1B*). As expected, tyrosine phosphorylation in WT BMDMs was insensitive to 3-IB-PP1. We conclude that the specific inhibitory effect of 3-IB-PP1 on Csk[AS], not a nonspecific effect, causes the measured signaling responses. Interestingly, we found that 3-IB-PP1 impaired Erk and Akt responses to M-CSF in Csk[AS] BMDMs, suggesting that the failure of Erk signaling in response to 3-IB-PP1 arises from an active signaling blockade rather than a quantitative failure of membrane-proximal signaling (*Figure 4—figure supplement 2*).

Treating Csk[AS] CD4/CD8 double-positive thymocytes with 3-IB-PP1 induces robust Erk phosphorylation only in the presence of small-molecule actin remodeling agents or when combined with costimulation through the CD28 pathway, which leads to actin remodeling (*Tan et al., 2014*). We performed analogous experiments with BMDMs by adding Cytochalasin D, Latrunculin A, or Jasplakinolide in combination with 3-IB-PP1 or zymosan[dep] (*Figure 4—figure supplement 3*). Although actin remodeling affected Erk phosphorylation in response to zymosan[dep], it did not increase Erk phosphorylation in response to 3-IB-PP1. This finding suggests that the signaling blockade between activated SFKs and Erk in the absence of receptor clustering is fundamentally different in macrophages than in thymocytes, reflecting the different functions of actin remodeling and costimulatory pathways in signal initiation.

# Priming macrophages with inflammatory cytokines overcomes their dependence on receptor clustering for downstream signaling

## Inflammatory priming of macrophages rescues signaling downstream of SFK activation by 3-IB-PP1

Macrophages become more responsive to some stimuli following priming with inflammatory cytokines such as interferon-γ (IFN-γ). To test the effects of inflammatory priming on SFK-mediated

signaling, we exposed BMDMs to IFN-γ for 12–16 hr prior to treatment with zymosan[dep] or 3-IB-PP1. Strikingly, priming with IFN-γ enabled 3-IB-PP1 treatment to stimulate robust downstream signaling through Erk (*Figure 5A*, arrow) as well as Akt, the MAPK JNK, and the calcium-sensitive transcription factor NFAT (*Figure 5B*, arrows). Priming with IFN-γ had a comparatively small effect on downstream signaling in response to zymosan[dep].

Priming BMDMs with IFN-γ generally increased membrane-proximal signaling in response to 3-IB-PP1. We observed more robust phosphorylation of DAP12, Syk, LAT, PLCγ1, PLCγ2, Cbl, PI3K, HS1, Vav, and Fak in primed cells than in unprimed cells (*Figure 5*, *Figure 5—figure supplement 1A*). Pulse-spinning alone did not induce robust Erk activation in primed cells (*Figure 5—figure supplement 1B*).

In addition to being more robust, phosphorylation of signaling proteins downstream of Syk (e.g., LAT, PLCγ1/2, c-Cbl, and PI3K) was more sustained in primed cells (*Figure 5*). This observation is

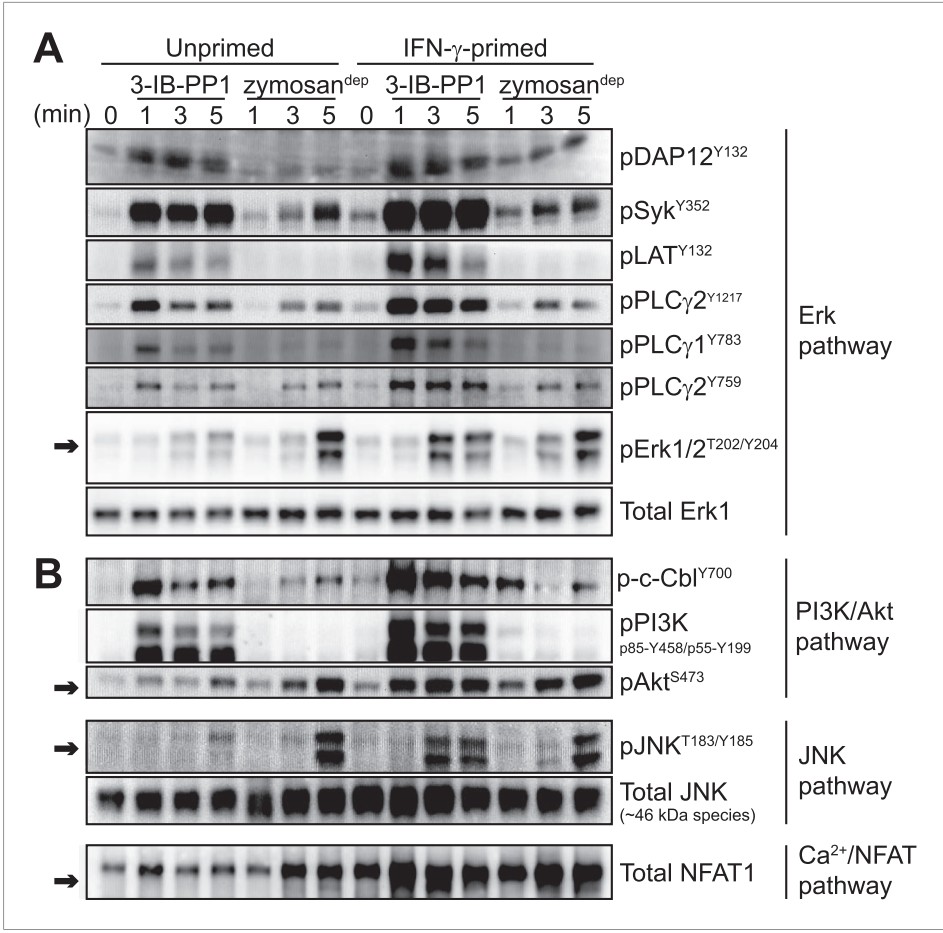

**Figure 5**. Inflammatory priming is required for SFK activation to produce downstream signaling in the absence of receptor clustering. (**A**) Csk[AS] BMDMs were incubated 12–16 hr in non-priming medium or in medium containing 25 U/ml IFN-γ. Signal transduction after 3-IB-PP1 or zymosan[dep] treatment was assessed by immunoblotting with antibodies specific to activating phosphorylation sites of DAP12, Syk, LAT, PLCγ2, PLCγ1, and Erk (arrow). Total Erk1 is shown as a loading control. (**B**) Signal transduction in the PI3K/Akt and JNK pathways was assessed by immunoblotting with antibodies specific for phosphorylated c-Cbl, PI3K, Akt (arrow), and JNK (arrow). Total JNK is shown as a loading control. NFAT1 activation was assessed by the faster migration of cellular NFAT1 upon dephosphorylation. See *Figure 5—figure supplement 1* for further characterization of signaling in primed and unprimed cells.

The following figure supplement is available for figure 5:

**Figure supplement 1**. Further characterization of signaling in primed and unprimed BMDMs.

especially striking because Syk phosphorylation continued to increase over time during treatment with 3-IB-PP1 in either primed or unprimed cells and also during treatment with zymosan[dep] and was therefore kinetically uncoupled from downstream signaling through Erk and Akt. Additionally, even the transient, low-level proximal signaling in response to 3-IB-PP1 in unprimed cells was quantitatively equivalent to or higher in magnitude than the signaling in response to zymosan[dep]. Finally, as mentioned above, cotreatment with 3-IB-PP1 suppressed signaling through the CSF-1R pathway (*Figure 4—figure supplement 2*). Together, these observations reinforce our claim that unprimed cells do not simply fail to reach a quantitative signaling threshold. Rather, a more qualitative difference in signaling circuitry explains the differential effects of priming on 3-IB-PP1 vs zymosan[dep] signaling. After observing a general enhancement of membrane-proximal signaling in primed cells, we hypothesized that priming directly influences the activation of all the membrane-proximal signaling proteins we tested.

We first considered the possibility that signaling proteins involved in negative feedback, themselves activated by the SFKs, are somehow bypassed in primed cells treated with 3-IB-PP1. Surprisingly, we found that priming actually enhanced the phosphorylation of proteins that negatively regulate signaling, including the ITIM-containing protein Sirpα, the inositol phosphatase SHIP1, and the tyrosine phosphatase SHP-1 (*Figure 6*), which led to activation of the negative-regulatory adaptor Dok-3 (*Figure 6—figure supplement 1*). These data indicate that priming does not generally inhibit negative feedback in macrophages. However, the increased phosphorylation of negative regulatory proteins in primed cells provided a clue to the upstream factor driving signaling in response to 3-IB-PP1: Since the SFK Lyn is the primary kinase responsible for phosphorylating and activating ITIMs and their associated phosphatases (*Ravetch and Lanier, 2000*; *Lowell, 2010*), we hypothesized that Lyn is driving signaling and that its expression or function is enhanced by IFN-γ priming.

## Activity of the SFK LynA is required for signaling through Erk and Akt in the absence of induced receptor clustering

### Inflammatory priming leads to Lyn upregulation

We wanted to understand whether IFN-γ enables Erk signaling in 3-IB-PP1-treated Csk[AS] BMDMs by means of direct cross-talk between IFN-γ and Erk signaling pathways or by a priming-mediated transcriptional change. We therefore treated BMDMs with IFN-γ in combination with cycloheximide to block translation, but we found that cycloheximide exposure was toxic and blocked all signaling through Erk. As an alternative, we tested different IFN-γ incubation times to look at the kinetics of the signaling enhancement. We found that 3-IB-PP1 failed to induce robust Erk phosphorylation in cells treated with IFN-γ for 4 hr or less (*Figure 7A*). IFN-γ priming for 6 hr or 8 hr produced modest Erk phosphorylation, and priming for 16 hr led to robust Erk phosphorylation in response to 3-IB-PP1. This temporal pattern strongly suggests that a priming-dependent transcriptional change potentiates Erk signaling after 3-IB-PP1 treatment.

Blotting for total Lyn protein in the presence and absence of IFN-γ revealed that Lyn expression was substantially increased by priming (*Figure 7A*, left lanes, *Figure 7—figure supplement 1*) and that Lyn expression correlated with the ability of 3-IB-PP1 to induce Erk signaling (*Figure 7A*, right lanes). We also tested an alternative priming agent, the growth factor GM-CSF (*Goodridge et al., 2009*), and found an intermediate degree of Lyn upregulation that corresponded with an intermediate level of Erk phosphorylation after 3-IB-PP1 treatment (*Figure 7B*). Moreover, among the SFKs and

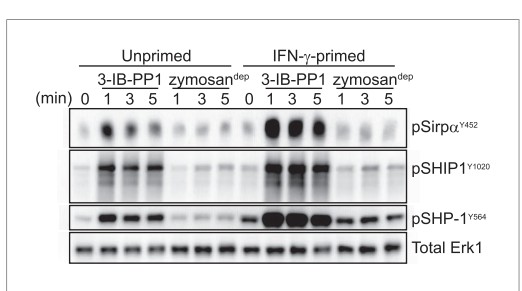

**Figure 6**. Activated SFKs initiate negative-regulatory ITIM pathways. Immunoblots show phosphorylation of negative regulatory proteins Sirpα, SHIP1, and SHP-1 in unprimed and IFN-γ-primed Csk[AS] BMDMs treated with 3-IB-PP1 or zymosan[dep]. See *Figure 6—figure supplement 1* for evidence that the negative-regulatory adaptor Dok-3 is also activated in response to 3-IB-PP1 treatment.

The following figure supplement is available for figure 6:

**Figure supplement 1**. Dok-3 activation by 3-IB-PP1.

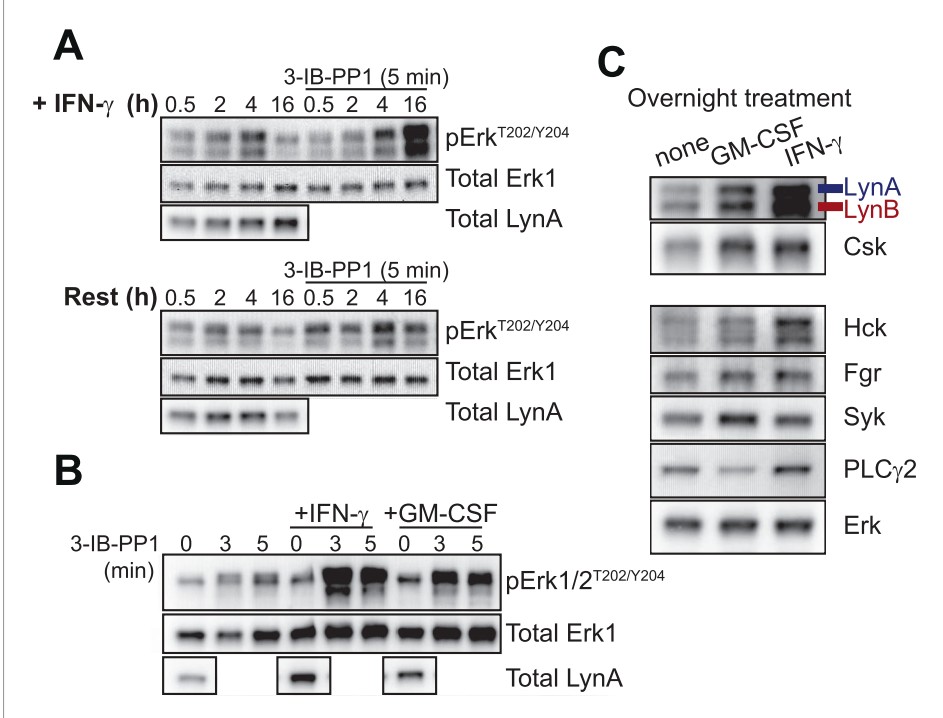

**Figure 7**. Lyn upregulation after priming coincides with Erk signaling after SFK activation. (**A**) Csk[AS] BMDMs were incubated in non-priming medium or in priming medium containing 25 U/ml IFN-γ and then analyzed directly or treated with 3-IB-PP1 for 5 min. Erk phosphorylation and Lyn expression were assessed by immunoblot. See *Figure 7—figure supplement 1* for uncropped LynA blots. (**B**) Csk[AS] BMDMs were primed overnight in 25 U/ml IFN-γ or 10 ng/ml GM-CSF. Erk phosphorylation after 3-IB-PP1 treatment and basal Lyn expression are shown. (**C**) Immunoblots show basal expression of Lyn, Hck, Fgr, Syk, and PLCγ2 with and without priming.

The following figure supplement is available for figure 7:

**Figure supplement 1**. Uncropped Lyn blots.

other membrane-proximal signaling proteins examined, only Lyn and Csk itself were upregulated in both GM-CSF-primed and IFN-γ-primed BMDMs (*Figure 7C*). This observation is consistent with raw data from a recent microarray study showing increased Lyn expression in IFN-γ-primed macrophages (*Noubade et al., 2014*). We therefore hypothesized that Lyn upregulation during inflammatory priming enables downstream signaling in BMDMs in response to 3-IB-PP1.

## Lyn expression is required for downstream signaling in response to 3-IB-PP1 but not receptor clustering

To confirm that Lyn expression is important for Erk signaling in response to 3-IB-PP1, we crossed *Csk[AS]* mice with Lyn-deficient mice (*Chan et al., 1997*) to generate *Lyn[−/−]Csk[AS]* mice. Strikingly, Lyn deficiency rendered BMDMs completely unresponsive to the signal-potentiating effects of priming; Lyn[−/−]Csk[AS] BMDMs primed with IFN-γ failed to signal through Erk after 3-IB-PP1 treatment (*Figure 8A*, arrow) even though they express normal macrophage surfaces markers (*Figure 8—figure supplement 1*). In contrast, the magnitude of downstream signaling in response to zymosan[dep] was unaffected by Lyn deficiency. This reveals a fundamental difference in the requirement for Lyn in the SFK-mediated signaling initiated by 3-IB-PP1 compared to receptor clustering. Hck and Fgr, the other major SFKs expressed in macrophages, cannot compensate for the loss of Lyn to promote downstream signaling during 3-IB-PP1 treatment. However, in ligand-induced receptor clustering Hck and Fgr do compensate for Lyn and downstream signaling is able to proceed.

When we treated Lyn[−/−]Csk[AS] BMDMs with 3-IB-PP1, the remaining SFKs, Hck and Fgr, induced the phosphorylation of Vav, Syk, LAT, PLCγ, and SHIP1 (*Figure 8A*). Although phosphorylation of these

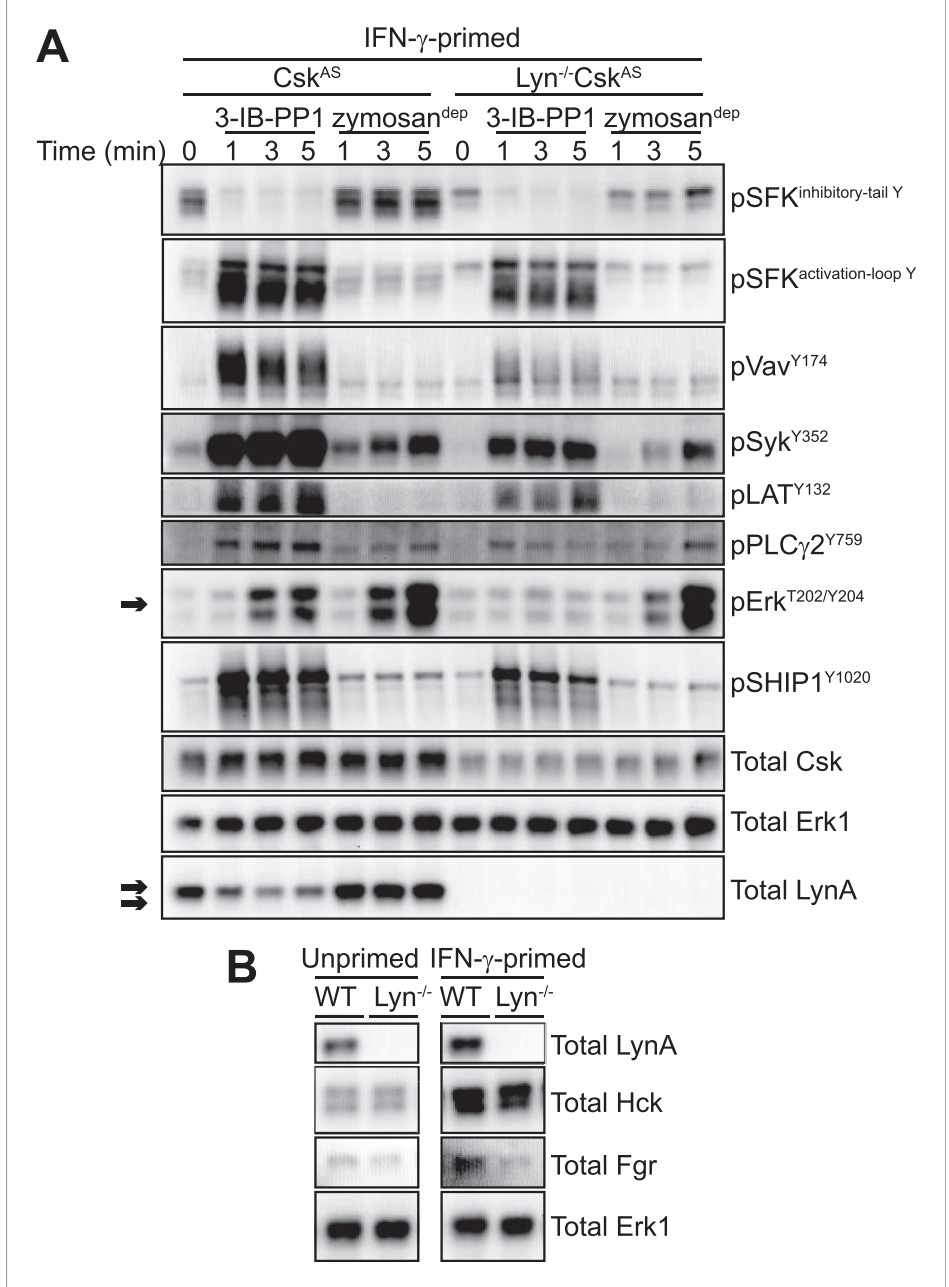

**Figure 8**. IFN-γ-potentiated signaling through Erk after Csk inhibition depends on Lyn expression. (**A**) Signaling in IFN-γ-primed CskAS and Lyn−/−CskAS BMDMs was assessed by immunoblotting as described previously. Erk phosphorylation is highlighted by a single arrow, total LynA level is highlighted by a double arrow. (**B**) Immunoblots show SFK expression in unprimed and IFN-γ-primed WT and Lyn−/− BMDMs. See *Figure 8—figure supplement 1* for evidence that Lyn-deficient BMDMs display normal surface markers.

The following figure supplement is available for figure 8:

**Figure supplement 1**. Lyn deficiency does not impair BMDM generation.

membrane-proximal signaling proteins was less robust in Lyn−/−CskAS BMDMs, the level of proximal signaling still surpassed the level detected after zymosandep treatment. We conclude that Lyn activity is specifically required for priming-enabled Erk signaling in response to 3-IB-PP1.

SFKs have some functional redundancy, and cells sometimes compensate for the loss of a single kinase with increased expression of the remaining SFKs (*Majeed et al., 2001*; *Luo et al., 2010*). No

changes in expression of the other major macrophage SFKs, Hck and Fgr, were observed in Lyn-deficient B cells (*Chan et al., 1997*), but we nevertheless tested Hck and Fgr expression in WT and Lyn$^{-/-}$ BMDMs. As expected, Hck and Fgr expression levels were not affected by Lyn expression (*Figure 8B*).

Hck is, however, upregulated in both WT and Lyn$^{-/-}$ BMDMs in response to IFN-γ priming (*Figure 8B*). Because it is not upregulated during GM-CSF priming and fails to rescue signaling in Lyn$^{-/-}$ cells, Hck is unlikely to be the dominant factor promoting signaling in response to 3-IB-PP1. However, the normal upregulation of Hck in response to IFN-γ indicates that Lyn$^{-/-}$ cells are not generally impaired in their response to IFN-γ. This supports our conclusion that some Lyn-specific function is responsible for downstream signaling in response to 3-IB-PP1 in IFN-γ primed cells.

The correlation between Lyn expression and Erk phosphorylation suggests that a threshold of Lyn abundance and activity determines whether SFK activation can induce Erk signaling in response to 3-IB-PP1. Interestingly, we found that the amount of Lyn protein decreased during the course of 3-IB-PP1 treatment (*Figure 8A*, double arrow), and this led us to quantify the levels of all of the SFKs during 3-IB-PP1 treatment.

## LynA is selectively degraded after Csk inhibition

We next examined the levels of Lyn, Hck, and Fgr protein during 3-IB-PP1-treatment of Csk$^{AS}$ BMDMs (*Figure 9A*). We assigned immunoblot bands to individual SFK species by testing antibodies against lysates from BMDMs derived from WT, Hck$^{-/-}$, and Lyn$^{-/-}$ mice (*Lowell et al., 1994*; *Chan et al., 1997*) and were able to differentiate between Hck, Fgr, and the two splice forms of Lyn, LynA (56 kDa) and LynB (53 kDa) (*Stanley et al., 1991*; *Yi et al., 1991*) (*Figure 9—figure supplement 1*). The two isoforms of Hck (59 kDa and 56 kDa) (*Lock et al., 1991*) were also distinguishable but behaved identically in our experiments and are not discussed separately.

We examined the levels of Lyn, Hck, and Fgr protein during 3-IB-PP1-treatment of Csk$^{AS}$ BMDMs and found that LynA was preferentially depleted (*Figure 9A*). Total LynA levels in unprimed cells treated with 3-IB-PP1 decreased by 60% within 2 min, whereas Hck, LynB, and Fgr each decreased by only about 30% during the same time frame (*Figure 9B*). We recapitulated this analysis using three independent Lyn antibodies, indicating that the disappearance of LynA from the immunoblots was unlikely to be caused by a posttranslational modification that blocked antibody recognition.

After treating Csk$^{AS}$ cells with 3-IB-PP1, the E3 ubiquitin ligase c-Cbl was robustly phosphorylated (*Figure 5B*), leading us to speculate that activated Lyn might be ubiquitinated as part of a c-Cbl-mediated negative feedback loop (*Rao et al., 2002*; *Kyo et al., 2003*). We used immunoprecipitation to isolate LynA and Hck in the presence or absence of 3-IB-PP1 and assessed the degree of ubiquitination of each immunoprecipitate by immunoblotting. We found that LynA immunoprecipitates were indeed more polyubiquitinated after treatment with 3-IB-PP1 (*Figure 9C*). To confirm that LynA itself was modified by ubiquitination, we also used immunoblotting to show laddering of LynA and activation-loop-phosphorylated SFKs in whole-cell lysates (*Figure 9—figure supplement 2*). Such laddering was not seen with Hck.

We were intrigued by the peculiar susceptibility of LynA to degradation, even compared to LynB. LynA differs from LynB only in a 21 amino-acid insert in its unique region, and this unique insert contains a predicted ubiquitination site (K40) and a phosphorylation site (Y32) (*Figure 9—figure supplement 3*) (*Stanley et al., 1991*; *Yi et al., 1991*; *Radivojac et al., 2010*; *Huang et al., 2013*). We therefore speculate that this insert regulates LynA uniquely and marks LynA preferentially for ubiquitination and degradation.

## Lyn is activated preferentially in response to 3-IB-PP1

Having established that LynA is selectively targeted for degradation, we looked more closely at the activation kinetics of the SFKs during the course of 3-IB-PP1 treatment (*Figure 10*). We identified the immunoblot bands that corresponded to inhibitory-tail-phosphorylated Lyn and Hck (*Figure 10—figure supplement 1A*); activation-loop-phosphorylated Hck, LynA, and LynB (*Figure 10—figure supplement 1B*); and activation-loop-phosphorylated Fgr (*Figure 10—figure supplement 1C*). We then analyzed the activation of each of the SFKs following 3-IB-PP1 treatment (*Figure 10A*).

Although both Lyn and Hck lost inhibitory-tail phosphorylation during 3-IB-PP1 treatment, the inhibitory tail of Lyn was dephosphorylated preferentially (5% Lyn vs 24% Hck tail phosphorylation

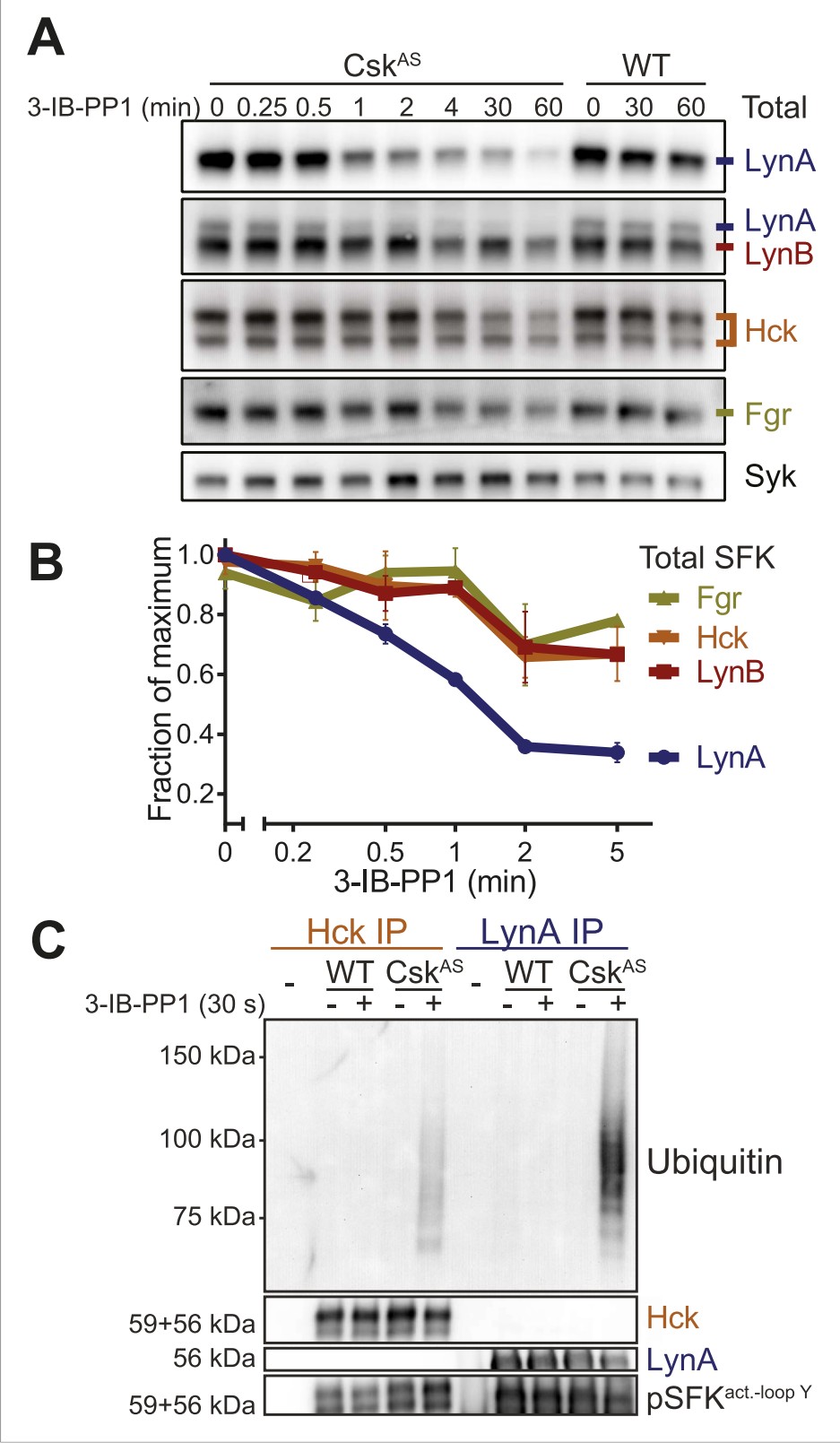

**Figure 9**. LynA is selectively degraded after Csk inhibition. (**A**) Csk[AS] and WT BMDMs were treated with 3-IB-PP1. Total kinase levels were detected by immunoblotting with antibodies to LynA, LynA+LynB, Hck, and Fgr. Total Syk is shown as a loading control. (**B**) SFK levels for the first 5 min of 3-IB-PP1 treatment were quantified by densitometry. *Figure 9. continued on next page*

*Figure 9. Continued*

Error bars reflect the standard deviation from three separate experiments. (**C**) Antibodies were used to immunoprecipitate Hck or LynA from BMDM lysates with or without treatment with 3-IB-PP1 for 30 s. The ubiquitination of each immunoprecipitate and the specificity and efficiency of the immunoprecipitation procedures were assessed by immunoblotting. See *Figure 9—figure supplement 1* for assignment of immunoblot bands, *Figure 9—figure supplement 2* for direct detection of LynA and activated SFK laddering in whole-cell lysate, and *Figure 9—figure supplement 3* for a sequence alignment of LynA and LynB that shows putative ubiquitination and phosphorylation sites unique to LynA.

The following figure supplements are available for figure 9:

**Figure supplement 1**. Identification of individual SFKs in immunoblots of whole-macrophage lysate.

**Figure supplement 2**. Laddering of activated Lyn in whole-cell lysate.

**Figure supplement 3**. Predicted ubiquitination site in the LynA insert.

remaining after 1 min) (*Figure 10B*). Similarly, we found that both Lyn isoforms were activation-loop phosphorylated more rapidly than Hck and Fgr. LynA and LynB reached maximum activation-loop phosphorylation within 3–6 s of 3-IB-PP1 addition, whereas activated Hck and Fgr continued to accumulate for the first 1–3 min (*Figure 10C*).

The active form of LynA was short-lived during 3-IB-PP1 treatment (*Figure 10C*), mirroring the pattern of total protein degradation. Only 20% of the activation-loop-phosphorylated LynA remained after 3 min, but the other activated SFKs, LynB, Hck, and Fgr persisted at 70%, 80%, and 90%, respectively, of the maximum level. All the SFKs were activated relative to their basal levels (*Figure 10—figure supplement 2*), but activated LynA was uniquely transient during 3-IB-PP1 treatment.

## Activated LynA persists in primed, 3-IB-PP1-treated BMDMs

Phosphorylation of Erk was detectable in primed BMDMs 3–5 min after zymosan[dep] or 3-IB-PP1 treatment. In unprimed BMDMs treated with 3-IB-PP1, LynA was almost completely degraded within this time frame. LynA protein was also subject to degradation in IFN-γ-primed cells during 3-IB-PP1 treatment, but due to its increased basal expression LynA protein persisted at significant levels through the 3- to 5-min time frame of Erk activation (*Figure 11A*, *Figure 11—figure supplement 1*). At each time point during the first 5 min of 3-IB-PP1 treatment, LynA levels were threefold higher in primed cells than in unprimed cells (*Figure 11B*). We conclude that the greater abundance of LynA protein in primed cells serves as a buffer against degradation, allowing LynA activity to be sustained above a threshold level and enabling Erk signaling in response to 3-IB-PP1.

Our data support a model in which the precise amount of LynA protein determines whether macrophages strictly require strong ITAM/hemi-ITAM receptor clustering for activation of MAPK, Akt, and calcium signaling pathways. The basal expression level of Lyn sets this threshold, and this level of expression can be tuned by the extracellular milieu: inflammation initiates a program that overrides this LynA-mediated signaling checkpoint, thereby sensitizing macrophages to stimuli that fail to cluster receptors efficiently. In addition, we have found that a unique function of receptor clustering reorganizes signaling through the SFKs, bypassing LynA and enabling Hck- and Fgr-mediated downstream signaling.

## Discussion

In this study, we examined signaling in BMDMs induced by activating the SFKs in the absence of receptor clustering (by inhibiting Csk[AS] with 3-IB-PP1) and compared this to signaling induced by receptor clustering (mediated by the Dectin-1 ligand zymosan[dep]). Treatment with 3-IB-PP1 induced robust membrane-proximal signaling, but signal propagation was blocked downstream of PLCγ and PI3K, with no apparent activation of Erk, JNK, Akt, or NFAT. We determined that propagation of receptor-independent signaling induced by 3-IB-PP1 requires the SFK LynA, but we observed that LynA protein was almost completely degraded within the first three minutes after 3-IB-PP1 addition.

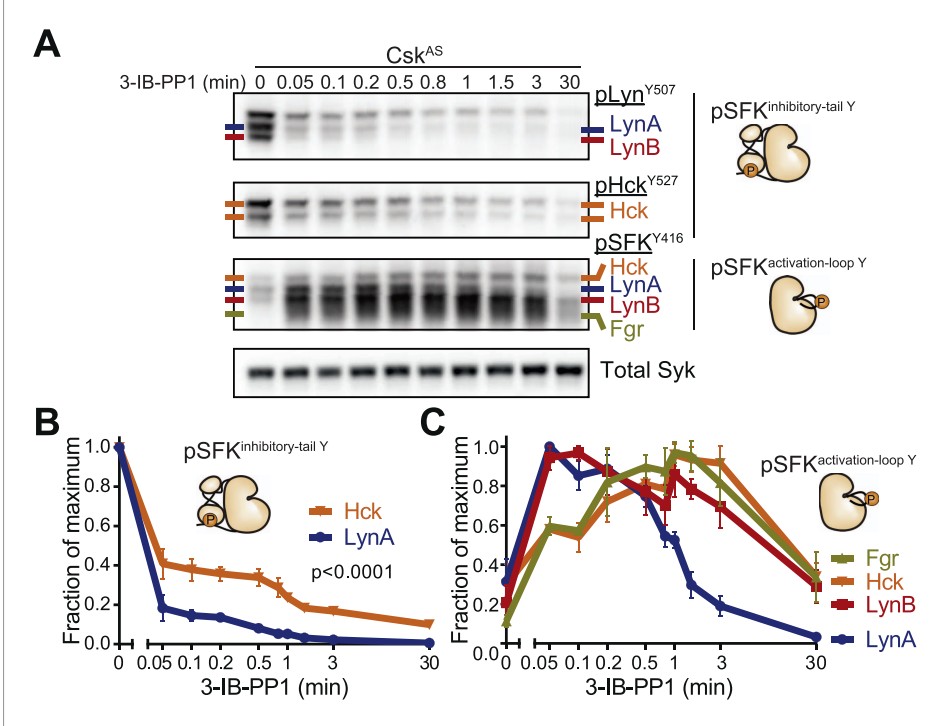

**Figure 10**. Csk inhibition by 3-IB-PP1 produces transient LynA activation. (**A**) Immunoblots of inactive and active SFKs in Csk[AS] BMDMs treated with 3-IB-PP1. Inactive Lyn and Hck were detected with the antibodies pLyn[Y507] and pHck[Y527], respectively. Active Lyn, Hck, and Fgr were detected with the pSFK[Y416] antibody. (**B**) Levels of inactive SFKs were quantified by densitometry. Error bars reflect the standard deviation from three separate experiments. p-values reflect two-tailed t tests. (**C**) Levels of active SFKs. A 2-way Anova shows significant pairwise differences between LynA and each other SFK species. See *Figure 10—figure supplement 1* for assignment of immunoblot bands and *Figure 10—figure supplement 2* for data rendered as fold increase in SFK activation with 3-IB-PP1 treatment.

The following figure supplements are available for figure 10:

**Figure supplement 1**. Identification of individual active and inactive SFKs in immunoblots of whole-macrophage lysate.

**Figure supplement 2**. Fold increase in SFK activation by 3-IB-PP1 treatment.

This blockade was overcome by priming macrophages with IFN-γ or GM-CSF before 3-IB-PP1 treatment, which led to increased expression of LynA, thus buffering it against total degradation and leading to a more sustained presence of LynA during the first minutes of 3-IB-PP1 treatment. The other macrophage SFKs, Hck and Fgr, were also activated in response to 3-IB-PP1 treatment but were not capable of compensating for the lack of LynA activity after LynA degradation or in Lyn-deficient cells. In contrast, receptor clustering by zymosan[dep] induced signaling through the MAPKs and Akt independently of LynA function and inflammatory priming.

3-IB-PP1 induces SFK activation without large-scale receptor clustering or formation of a phagocytic synapse, a quality shared with weak physiological stimuli. In the experiments discussed here, we treated cells with a high dose of 3-IB-PP1 to induce robust SFK activation and unmask signaling that would otherwise be undetectable with weak stimulation. When we titrate 3-IB-PP1 to lower doses, the signaling pattern is unchanged, just generally weaker in strength. Therefore, we believe that the strong SFK activation in response to 3-IB-PP1 is an advantage of this experimental system and allows the comparison of signaling in the absence of receptor clustering, as in a weak physiological stimulus, to signaling in response to zymosan[dep], which mimics a strong physiological stimulus (*Goodridge et al., 2011*). In this context, differences in signaling between 3-IB-PP1- and

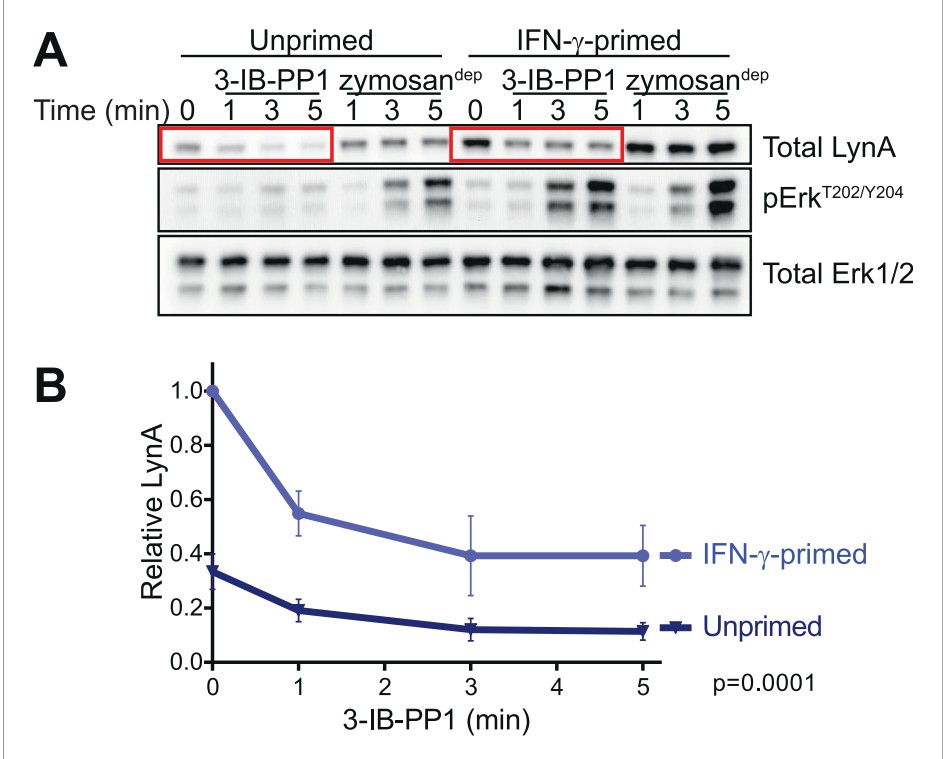

**Figure 11**. LynA persists in primed cells treated with 3-IB-PP1. (**A**) Immunoblots from unprimed and IFN-γ-primed Csk[AS] BMDM lysates show levels of LynA and corresponding Erk phosphorylation during 3-IB-PP1 treatment. (**B**) Kinetics of LynA degradation relative to the maximum value within each experiment (primed cells, t = 0). (Quantified by densitometry. Error bars reflect the standard deviation over four independent experiments. The p-value is derived from a ratio paired, two-tailed t test.) See *Figure 11—figure supplement 1* for independently normalized comparison of Lyn degradation in primed and unprimed cells.

The following figure supplement is available for figure 11:

**Figure supplement 1**. Priming does not inhibit LynA degradation.

zymosan[dep]-treated cells reveal how receptor clustering relays signaling information. Importantly, we found that Erk, JNK, Akt, and NFAT activation patterns do not only depend on the strength of SFK activation and membrane-proximal signaling, but instead arise from the different roles of LynA and the other SFKs in the presence and absence of induced receptor clustering. We conclude, therefore, that large-scale receptor clustering does qualitatively change the chain of events necessary for signaling to occur independently of general SFK activation strength.

We propose that the responsiveness of macrophages to weak stimuli is regulated by a signaling checkpoint in which degradation of LynA occurs when weak or low-valency ligands engage surface receptors; by contrast, LynA expression is maintained but not required when strong or high-valency ligands induce large-scale receptor clustering. As a result, signaling induced in the absence of significant receptor clustering is blocked at the level of Erk, JNK, Akt, and NFAT, while signaling induced by high-valency receptor clustering proceeds through these pathways. However, when cells are primed with IFN-γ, they upregulate expression of LynA sufficiently that even weak or low-valency ligands can produce downstream Erk, JNK, Akt, and NFAT signaling. While priming sensitizes macrophages to small amounts of high-valency pathogen ligands, it will also increase the chance to signaling to host-derived or nonpathogenic low-valency ligands that may lead to immunopathology.

Our finding that LynA promotes signaling in response to 3-IB-PP1 was initially surprising because Lyn-deficient myeloid cells have defects in negative regulatory signaling (*Harder et al., 2001*, *2004*; *Lamagna et al., 2013*). Lyn phosphorylates ITIMs and their associated phosphatases, and so Lyn

deficiency leads to hyperactivation of cell signaling (*Scapini et al., 2009*; *Ingley, 2012*). However, such Lyn-deficient cells lack both LynA and LynB. When mast cells derived from *Lyn*⁻/⁻ mice are reconstituted either with LynA or LynB, LynA expression leads to more robust PLCγ phosphorylation and better association of PLCγ with the adaptor protein LAT in response to FcεRI ligation. In contrast, LynB expression drives the formation of a LynB-SHIP1 complex and impairs calcium and degranulation responses (*Alvarez-Errico et al., 2010*). These observations are consistent with our conclusion that in macrophages, LynA is the primary mediator of downstream signaling in response to 3-IB-PP1. The persisting activity of LynB after LynA degradation is likely to activate negative feedback responses and ensure deterioration of membrane-proximal signaling. It is likely, however, that the loss of some positive function of LynA leads to the primary blockade in signaling because Lyn-deficient macrophages (lacking both LynA and LynB) lose all capacity to signal through Erk and Akt in response to 3-IB-PP1. In addition, LynB, like LynA, is upregulated during priming and fails to suppress LynA-mediated signaling. Therefore, we conclude that the effects of LynA activity are dominant or orthogonal to any negative-regulatory effects of LynB.

We do not yet know how LynA uniquely promotes downstream signaling, but we speculate that some activity of LynA may protect PLCγ from inhibition by Dok-3. Studies in B cells have shown that a complex of Dok-3, SHIP1, and Grb2 associates with Slp76, Btk, and PLCγ and negatively regulates PLCγ, Ras/MAPK, and Akt function (*Honma et al., 2006*; *Stork et al., 2007*; *Losing et al., 2012*). Two phosphorylation sites of Dok-3 (Y378 and Y399) are not required for SHIP1-binding or inhibitory function (*Lemay et al., 2000*; *Robson et al., 2004*), and we speculate that these sites could sense LynA activity and block Dok-3 function. We have detected general tyrosine phosphorylation of Dok-3 and its association with SHIP1 in response to 3-IB-PP1 (*Figure 6—figure supplement 1*), and the role of Dok-3 in signaling will be the subject of future studies.

Our observations diverge from the standard model of how SFK activation is reversed, namely through dephosphorylation of the activation loop by PTPN22/Pep or other phosphatases and subsequent phosphorylation of the inhibitory tail by Csk (*Brown and Cooper, 1996*; *Wu et al., 2006*; *Zikherman et al., 2010*; *Negro et al., 2012*; *Okada, 2012*). However, previous work has shown that activated SFKs can phosphorylate the E3 ubiquitin ligase c-Cbl, which then targets them for degradation (*Rao et al., 2002*). In fact, Lyn ubiquitination in mast cells has been observed within 15–30 s of FcεRI activation (*Kyo et al., 2003*). What surprised us was that bulk protein degradation appeared to account for all loss of activated Lyn after 3-IB-PP1 treatment. This mechanism is reminiscent of the regulation of receptor tyrosine kinases by Cbl (*Thien and Langdon, 2001*) and internalization/degradation processes for deactivating the T-cell receptor (*Liu et al., 2000*). This observation has important implications for how macrophages might respond to repeated encounters with weak stimuli. The LynA checkpoint would not only shut signaling down but also render macrophages refractory to restimulation by another weak stimulus, at least until redistribution at the membrane or new synthesis replenishes local reserves of LynA. We have observed one form of this desensitization during 3-IB-PP1 wash-out and subsequent stimulation with the growth factor M-CSF, but in vivo this process likely occurs in a highly spatially localized manner. This degradation and desensitization could prevent the accumulation of hyperactivated macrophages at sites rich in weak stimuli but lacking genuine pathogenic invaders.

Activation of macrophage subsets is a key factor in the progression and severity of autoimmune diseases, cancer cell growth and differentiation, the success of cancer immunotherapies, and induction and maintenance of obesity (*Wynn et al., 2013*). Moreover, alterations in macrophage sensitivity through defects in membrane-proximal negative regulators have already been linked directly to disease. For example, loss of negative-regulatory receptor tyrosine kinases Tyro3, Axl, and Mer exacerbates lupus-like disease, rheumatoid arthritis, and inflammatory bowel disease through enhanced Toll-like receptor signaling (*Rothlin et al., 2007*; *Lemke and Rothlin, 2008*; *Rothlin and Lemke, 2010*). Loss of either SHP-1 or SHIP1, both negative regulators of ITAM signaling, causes severe myeloproliferation, skin abscesses, and lung failure (*Van Zant and Shultz, 1989*; *Shultz et al., 1993*; *Tsui et al., 1993*; *Helgason et al., 1998*; *Abram et al., 2013*). The LynA checkpoint constitutes an additional system by which macrophages sense inflammation and set basal sensitivity accordingly. Understanding the ways that macrophages become hypersensitive to stimulation, including the bypass of the LynA checkpoint, will provide insights into how we might limit this process and reduce the contribution of macrophage signaling to disease pathology.

## Materials and methods

### Mice

All mice used in these experiments were derived from the strain C57/BL6. $Csk^{AS}$ mice (*Tan et al., 2014*) are heterozygous for a BAC transgenic $Csk^{AS}$ allele on a Csk-null background. $Hck^{-/-}$ mice (*Lowell et al., 1994*) and $Lyn^{-/-}$ mice (*Chan et al., 1997*) were used for antibody characterization and breeding. $Lyn^{-/-}Csk^{AS}$ mice were generated by crossing $Csk^{AS}$ and $Lyn^{-/-}$ mice. All animals were backcrossed to C57/BL6 for 15 generations.

### Preparation of BMDMs

BMDMs were prepared using standard methods (e.g., as described in *Zhu et al., 2008*). Bone marrow was extracted from femura/tibiae of mice. After hypotonic lysis of erythrocytes, BMDMs were derived on untreated plastic plates (BD Falcon, Bedford, MA) by culturing in Dulbecco's Modified Eagle Medium (Corning Cellgro, Manassas, VA) containing approximately 10% heat-inactivated fetal calf serum (Omega Scientific, Tarzana, CA), 0.11 mg/ml sodium pyruvate (UCSF Cell Culture Facility), 2 mM penicillin/streptomycin/L-glutamine (Sigma-Aldrich, St. Louis, MO), and 10% CMG-12-14-cell-conditioned medium as a source of M-CSF (*Takeshita et al., 2000*). After 6 or 7 days, cells were resuspended in enzyme-free ethylenediaminetetraacetic acid (EDTA) buffer and replated in untreated 6-well plates (BD Falcon) at 1 M cells per well in unconditioned medium ±25 U/ml IFN-γ (Peprotech, Rocky Hill, NJ) or 10 ng/ml GM-CSF (eBioscience, San Diego, CA; *Goodridge et al., 2009*).

### Surface staining of BMDMs

BMDMs were resuspended and stained with PE-Cy7-conjugated anti-CD11c (eBioscience clone N418, 25-0114), FITC-conjugated anti-F4/80 (eBioscience clone BM8, 11-4801), and PE-conjugated CD11b (BD Pharmingen, San Jose, CA, clone M1/70 553311). BD and eBioscience antibodies to CD45 were used as single-stained controls. Data were collected on a BD LSRFortessa flow cytometer running FACSDiva software and analyzed in FlowJo (TreeStar, Ashland, OR). Figures were made in Adobe Creative Suite (San Jose, CA).

### Preparation of depleted zymosan

Zymosan (Sigma) was depleted of TLR2 agonist as described previously (*Underhill, 2003*). Briefly, intact zymosan suspended in phosphate-buffered saline (PBS) was subjected to five 15-min cycles of boiling and sonication followed by pelleting and resuspension in fresh PBS. The sample was then boiled in 10 M NaOH for 1 hr, washed, resuspended in PBS, and counted. Stocks of 1000 M zymosan^dep particles per ml were stored at −20°C and thawed, briefly sonicated, pelleted, and resuspended before use.

### Macrophage stimulation

Stimuli prepared in Roswell Park Memorial Institute- (RPMI-)1640 medium were added to adherent BMDMs at 37°C. In experiments with zymosan^dep, all stimuli were applied by pulse spinning at 37°C. BMDMs were stimulated with 10 zymosan^dep particles per cell (*Underhill, 2003*), 10 μM 3-IB-PP1 $Csk^{AS}$ inhibitor (*Okuzumi et al., 2009*; *Tan et al., 2014*), 2–50 ng/ml recombinant mouse M-CSF (eBioscience), 1 μM BAY 61-3606 Syk inhibitor (Calbiochem; *Yamamoto et al., 2003*), 20 μM PP2 SFK inhibitor (Calbiochem EMD Millipore, Billerica, MA; *Hanke et al., 1996*) or the actin-remodeling agents Cytochalasin D (10 μM), Latrunculin A (0.5 μM), or Jasplakinolide (1 μM). Reactions were stopped by placing the plate on ice, aspirating the stimulus, adding nonreducing SDS sample buffer, scraping cells off the plate, incubating 5 min at 37°C, adding 50 mM dithiothreitol (DTT), sonicating on a Diagenode Biorupter (Denvillle, NJ), and boiling for 15 min. Samples were separated for immunoblot analysis by sodium dodecyl sulfate polyacrylamide gel electrophoresis (SDS-PAGE) (Novex NuPAGE Thermo Fisher Scientific, Grand Island, NY).

### Immunoprecipitation

Stimulated BMDMs were lysed in 1% Lauryl Maltoside Lysis Buffer containing 150 mM NaCl, 0.01% sodium azide, and 10 mM Tris, pH 7.6 with 2 mM $NaVO_4$, 0.01 mg/ml Aprotinin, 0.01 M NaF, 0.01 mg/ml Leupeptin, 0.01 mg/ml Pepstatin A, 2 mM phenylmethanesulfonyl fluoride (PMSF), and 0.4 mM

EDTA. After scraping the plates, cells and detergent were incubated 30 min on ice. The lysate was then cleared by ultracentrifugation for 15 min at 50,000 rpm at 4°C in a Beckman (Pasadena, CA) TLA120.2 rotor. The lysates were precleared for 30 min at 4°C with Protein G Sepharose beads (Life Technologies, Carlsbad, CA) and normal rabbit or goat serum as appropriate (Jackson Immuno-Research, West Grove, PA). Protein G-Sepharose beads were covalently conjugated to Lyn or Hck antibodies (see table below) using dimethyl pimelimidate (Sigma). Antibody-bound beads were added to the lysate and mixed 1.5 hr at 4°C to immunoprecipitate Hck or Lyn. Finally, the samples were applied to micro bio-spin chromatography columns (Bio-Rad, Hercules, CA), washed, and eluted with SDS Sample Buffer.

## Antibodies and immunoblotting

After transfer from SDS-PAGE to Immobilon P membrane (EMD Millipore), blots were blocked with 3% bovine serum albumin (BSA) in 25 mM Tris, pH 8.0; 125 mM NaCl; and 0.02% $NaN_3$. Antibodies were applied in solutions containing 20 mM Tris, pH 8.0; 125 mM NaCl; and 0.05% Tween-20 (plus 2% BSA for primary or 0.5% powered milk for secondary antibody solution). High-stringency washes after antibody binding were in similar Tris-buffered saline and Tween-20 (TBST) buffer with 200 mM NaCl. Antibodies were obtained from Cell Signaling Technology (Danvers, MA), Santa Cruz (Dallas, TX), ProMab Biotechnologies (Richmond, CA), Promega (Madison, WI), Life Technologies, and Sigma-Aldrich. Primary staining was performed with the following antibodies:

| Antibody | Source-catalog no. | ID/clone |
|---|---|---|
| Erk1 | Santa Cruz-93-G | C-16-G |
| Erk1/2 | Santa Cruz-93 + Santa Cruz-154 | C-16 + C-14 |
| Fgr | ProMab-20318/Santa Cruz-50338 | 6G2 (PM)/M-60 (SC) |
| Hck | Santa Cruz-1428 | M-28 |
| JNK | Cell Signaling-9252 | – |
| LynA | Cell Signaling-2796/Lowell Lab | C13F9 (CS)/7478.5 (Lowell) |
| LynA + B | Santa Cruz-15 | 44 |
| NFAT1 | Cell Signaling-5861 | D43B1 |
| $pAkt^{S473}$ | Cell Signaling-4058 | 244F9 |
| $p\text{-}c\text{-}Cbl^{Y700}$ | Cell Signaling-8869 | D16D7 |
| $pErk1/2^{T202/Y204}$ | Cell Signaling-4377 | 197G2 |
| $pJNK1/2^{T183/Y185}$ | Promega-V7931 | – |
| $pLyn^{Y507}$ | Cell Signaling-2731 | – |
| $pHck^{Y527}$ | Life-44-912 | Anti-pSFK^neg |
| $pPI3K^{p85\text{-}Y458/p55\text{-}Y199}$ | Cell Signaling-4228 | – |
| $pPLC\gamma1^{Y783}$ | Life-44-696 G | – |
| $pPLC\gamma2^{Y759}$ | Cell Signaling-3874 | – |
| $pPLC\gamma2^{Y1217}$ | Cell Signaling-3871 | – |
| $pSFK^{Y416}$ | Cell Signaling-4058 | – |
| $pSHIP1^{Y1020}$ | Cell Signaling-3941 | – |
| $pSHP\text{-}1^{Y564}$ | Cell Signaling-8849 | D11G5 |
| $pSirp\alpha^{Y452}$ | Ken Swanson Lab | 10-1989 |
| $pSyk^{Y352}$ | Cell Signaling-2717 | 65E4 |
| pTyrosine | Weiss Lab + Millipore-05-947 | 4G10 + pY20 |
| Syk | Weiss Lab | 5F5 |
| SHIP1 | Santa Cruz-8425 | P1C1 |
| Vinculin | Sigma-Aldrich-V9264 | hVIN-1 |

Horseradish peroxidase- (HRP) conjugated secondary antibodies were from Southern Biotech (Birmingham, AL), and blots were developed using Thermo Scientific SuperSignal Femto reagent. Blots were visualized on a Kodak Imagestation. Some blots were reprobed after HRP inactivation by azide treatment and freezing.

### Kinetic analysis

Where applicable, immunoblots were analyzed quantitatively by performing densitometry with ImageJ software (Bethesda, MD). Briefly, images were background-subtracted, and strips of the blot corresponding to each band were demarcated and analyzed for each time point/gel lane. Error bars reflect independent analysis of blots from at least three independent experiments. Figures were prepared in Graphpad Prism (La Jolla, CA).

## Acknowledgements

Thanks to Kevan Shokat, David Underhill, Clare Abram, Haopeng Wang, Terri Kadlecek, Henrik Flach, Wan-Lin Lo, Erik Peterson, and Nicholas Levinson for advice, discussion, and/or critical reading of the manuscript. This work was supported primarily by NIH grants F32AI082926, 5T32AR007304-33, and 5T32AR007304-34 (TSF); 5P01AI091580 (AW); and RO1AI65495, RO1AI68150 and RO1AI113272 (CAL). T.S.F. was also supported by University of Minnesota Foundation Award NF-0315-02.

## Additional information

### Funding

| Funder | Grant reference | Author |
|---|---|---|
| National Institutes of Health | F32AI082926 | Tanya S Freedman |
| University of Minnesota Foundation | Award NF-0315-02 | Tanya S Freedman |
| National Institutes of Health | 5T32AR007304-33 | Tanya S Freedman |
| National Institutes of Health | 5T32AR007304-34 | Tanya S Freedman |
| National Institutes of Health | 5P01AI091580 | Arthur Weiss |
| National Institutes of Health | RO1AI65495 | Clifford A Lowell |
| National Institutes of Health | RO1AI68150 | Clifford A Lowell |
| National Institutes of Health | RO1AI113272 | Clifford A Lowell |

The funders had no role in study design, data collection and interpretation, or the decision to submit the work for publication.

### Author contributions

TSF, Conception and design, Acquisition of data, Analysis and interpretation of data, Drafting or revising the article, Contributed unpublished essential data or reagents; YXT, Acquisition of data, Analysis and interpretation of data, Drafting or revising the article; KMS, BNM, Acquisition of data, Drafting or revising the article, Contributed unpublished essential data or reagents; FVS, Acquisition of data, Analysis and interpretation of data, Drafting or revising the article, Contributed unpublished essential data or reagents; HSG, Analysis and interpretation of data, Drafting or revising the article, Contributed unpublished essential data or reagents; CAL, Conception and design, Analysis and interpretation of data, Drafting or revising the article; AW, Conception and design, Analysis and interpretation of data, Drafting or revising the article, Contributed unpublished essential data or reagents

### Ethics

Animal experimentation: This study was performed in strict accordance with the recommendations in the Guide for the Care and Use of Laboratory Animals of the National Institutes of Health. All of the animals were handled according to approved Institutional Animal Care and Use Committee (IACUC) protocol (AN107127-01E) of the University of California at San Francisco. Every effort was made to minimize suffering.

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
