## [Decision Letter]

Thank you for submitting your work entitled “LynA regulates an inflammation-sensitive signaling checkpoint in macrophages” for peer review at *eLife*. Your submission has been favorably evaluated by Tadatsugu Taniguchi (Senior editor) and three reviewers, one of whom is a member of our Board of Reviewing Editors.

The reviewers have discussed the reviews with one another and the Reviewing editor has drafted this decision to help you prepare a revised submission.

This manuscript reports an interesting phenomenon associated with macrophage priming. Using a technique that allows them to pharmacologically inhibit the CSK kinase, they report rapid activation of SFK activity and robust phosphorylation of immediate downstream effectors but inability to activate ERK, suggesting some type of block. This is in contrast to activation by zymosan, which achieves similar levels of phosphorylated downstream intermediates but also allows for strong ERK activation. The ability of CSK inhibition to activate ERK could be overcome by macrophage priming suggesting that priming allows for some rewiring. Analysis of the mechanism revealed that CSK inhibition activates SFK and results in preferential inhibition of the LynA isoform. Presumably, the loss of LynA prevents a sustained enough of a signal to allow for ERK activation.

There are many important findings in this manuscript. First, the data clearly show how dynamic the tyrosine phosphorylation of the SFK is as changes in phosphorylation are seen after 3 seconds. Second, the data show fundamental difference in LynA vs LynB, suggesting that LynA is responsible for positive signaling while LynB is responsible for negative regulator signaling. Third, it is shown that LynA is preferentially phosphorylated by CSK compared to other SFK and that LynA is also preferentially degraded. Fourth, the manuscript shows how rapidly SFK are lost after their activation. This is a little studied but important area. Lastly, the manuscript shows that an important effect of priming is up regulation of Lyn expression.

The manuscript could be improved by addressing several issues that are raised by the data:

1) The LynA degradation mechanism controls macrophage activation in response to pharmacological Csk inhibition, but does it also apply in response to a physiological stimulus?

2) It is stated that the increased expression of LynA after priming results in delayed degradation. Is it really delayed degradation or that the larger pool is able to sustained signaling longer because it takes longer to degrade to the threshold level.

3) It is proposed that LynA has an insert domain that contains a potential ubiquitination site and that this may account for the increased degradation of LynA compared to LynB. Is there any evidence to support this? Moreover, the authors point to a potential site of ubiquitination in this insert domain – is this site functionally relevant? Would a K40 mutant prevent LynA ubiquitination and degradation, and rescue activation of Erk and Akt.

4) Are both LynA and B unregulated after priming? If so, why doesn't the increased expression of LynB attenuate or block the increased signaling mediated by LynA?

5) If the inability of CSK inhibition to activate ERK is via increased degradation of LynA, it might be predicted that the kinetics of phosphorylation of immediate downstream substrates might be more transient. Is it?

6) The authors need to discuss the possibility that the effect of interferon-γ is simply quantitative. Is it possible that the ability of the cytokine to enable some activation of Erk and Akt is simply due to increased tyrosine phosphorylation?

---

## [Author Response]

The manuscript could be improved by addressing several issues that are raised by the data:

1) The LynA degradation mechanism controls macrophage activation in response to pharmacological Csk inhibition, but does it also apply in response to a physiological stimulus?

To mimic a soluble, physiological stimulus like cell debris, we treated BMDMs with laminarin, a soluble form of β-glucan that interacts with Dectin-1 but forms sub-threshold-sized receptor clusters (12). These new data show consistently that priming WT BMDMs with IFN-γ increases Erk phosphorylation induced by treatment with laminarin, in contrast to our observations with zymosan^dep^. Representative data are shown in the Figure 12.

Author response image 1.**DOI:**
http://dx.doi.org/10.7554/eLife.09183.030

Unfortunately, Dectin-1 ligation failed to produce sufficient SFK activation to assess whether local LynA ubiquitination and degradation block signaling as we predict, and explain the suggestive data above. Although SFK activity is essential for signaling in response to zymosan^dep^ (shown in Figure 3 of the manuscript), we cannot visualize bulk activation of the SFKs in response to this stimulus (shown in the pSFKY416 blot in Figure 4 of the manuscript).

To confirm this observation with zymosan^dep^ and laminarin treatment in parallel, we performed immunoprecipitation experiments with antibodies against LynA or the activated SFKs and blotted for ubiquitination and total LynA/activation-loop-phosphorylated SFK levels in the immunoprecipitates and the lysates. These new data are shown in Figure 13.

Author response image 2.**DOI:**
http://dx.doi.org/10.7554/eLife.09183.031

We do not yet know whether our inability to detect SFK activation in response to Dectin-1 ligation is because a pool of basally active SFKs that is harnessed for receptor signaling is too small or whether SFK activation is too highly localized to be detected in whole-cell lysate. Recent studies of the T-cell SFKs have implicated a basally active pool of Lck as the major driver of TCR signaling (33; 37). On the other hand, previous microscopy studies have shown that a pool of activated SFKs may be enriched within the phagocytic synapse (12). Please see paragraph four of the subsection “SFK activation in the absence of receptor clustering fails to induce downstream signaling” for a brief discussion on this observation.

Regardless of the cause, lack of bulk SFK activation in response to physiological stimuli has interfered with our detection of Lyn degradation. This difficulty of interpreting data from highly localized stimuli is precisely the reason that the Csk^AS^ system is so powerful: the bulk SFK activation permitted better detection of signaling events and their downstream outcomes.

We do agree that a long-term goal of our ongoing work should be to investigate the nanoscale SFK environment around ligated receptors. However, we will need to develop new reagents and seek a collaboration to perform high-resolution microscopy, which we believe lies beyond the scope of this paper and the appropriate time-frame for resubmission.

2) It is stated that the increased expression of LynA after priming results in delayed degradation. Is it really delayed degradation or that the larger pool is able to sustained signaling longer because it takes longer to degrade to the threshold level.

Thank you for pointing out that our argument was not clear in the initial manuscript. We were actually arguing that the larger pool of LynA is able to sustain signaling longer because it takes longer to degrade to the threshold level. We have updated the manuscript to resolve this confusion. Please refer to the last paragraphs of the Results and first paragraph of the Discussion.

3) It is proposed that LynA has an insert domain that contains a potential ubiquitination site and that this may account for the increased degradation of LynA compared to LynB. Is there any evidence to support this? Moreover, the authors point to a potential site of ubiquitination in this insert domain – is this site functionally relevant? Would a K40 mutant prevent LynA ubiquitination and degradation, and rescue activation of Erk and Akt.

We have strengthened our evidence for direct modification of LynA by ubiquitination (in contrast to Hck and the inactive forms of the SFKs) by using immunoblotting to detect laddering of LynA and activation-loop-phosphorylated SFKs in whole-cell lysates from Csk^AS^BMDMs treated with 3-IB-PP1. These data have been added as a new figure supplement (Figure 9—figure supplement 2). Please see the third paragraph of the subsection “LynA is selectively degraded after Csk inhibition”.

We do not yet have direct evidence that ubiquitination at the unique residue K40 targets LynA specifically for degradation, although we mention that this is the only lysine residue present in LynA and absent in LynB.

We attempted to obtain experimental support for our model by infecting mouse bone marrow with lentivirus containing LynA, LynAK40R, Hck, or LacZ control DNA. We expected to observe enhanced signaling in response to 3-IB-PP1 in cells overexpressing LynA or expressing any LynAK40R, which lacks the uniquely predicted ubiquitination site. In fact, we found that the introduction of LynA or LynAK40R into WT or even Lyn^-/-^ cells was toxic, whereas cells transduced instead with LacZ or Hck survived blasticidin selection. Although we were disappointed that we encountered this technical limitation, this observation of differential survival does support our model, since enhancing Lyn activity appears to affect the basal state of macrophages but enhancing Hck activity does not.

We are planning future experiments involving mass spectrometry, mutation of the other ubiquitination sites of Lyn, and using CRISPR-Cas methodology to knock the Lyn mutation into macrophages while maintaining physiological expression levels. These lines of inquiry, however, lie beyond the scope of this study and the appropriate timeframe for resubmission.

4) Are both LynA and B unregulated after priming? If so, why doesn't the increased expression of LynB attenuate or block the increased signaling mediated by LynA?

LynA and LynB are indeed both upregulated equally during priming, as we show in Figure 7. From our observation that LynB does not override LynA function, we argue that the effect of LynA is dominant (i.e. some positive function of LynA is more consequential than (or orthogonal to) any negative-regulatory signaling by LynB.) We have clarified this point in the manuscript (Discussion). Future studies will focus on investigating the mechanism of this signaling interplay.

5) If the inability of CSK inhibition to activate ERK is via increased degradation of LynA, it might be predicted that the kinetics of phosphorylation of immediate downstream substrates might be more transient. Is it?

We appreciate the reviewers' observation and have amended the manuscript to point this out. Indeed (as shown in Figure 5), membrane-proximal signaling in response to 3-IB-PP1 is more transient downstream of Syk in unprimed cells and more sustained in primed cells, consistent with our model that sustained LynA activity is necessary to promote signaling continuously to generate an Erk response. This observation is complicated by the fact that Syk phosphorylation itself is not transient in either case, rather activated Syk continues to accumulate well beyond the point of Erk signaling initiation. Also, the quantitative amount of even the transient signaling in unprimed cells is greater with 3-IB-PP1 than with zymosan^dep^, so quantity of signaling cannot be the complete story. Please see the subsection “Priming macrophages with inflammatory cytokines overcomes their dependence on receptor clustering for downstream signaling”.

*6) The authors need to discuss the possibility that the effect of interferon-*γ
*is simply quantitative. Is it possible that the ability of the cytokine to enable some activation of Erk and Akt is simply due to increased tyrosine phosphorylation?*

This comment is closely related to our discussion above of major point 5. The reviewers are correct that there are quantitative differences in signaling in primed vs. unprimed cells, depending on the level of LynA persisting during the first 5 min of 3-IB-PP1 treatment (Figure 5). The qualitative differences, we argue, are in the requirement for Lyn protein in the absence of receptor clustering vs. the lifting of this requirement in the context of receptor clusters (Figure 8). Again, we clarified the manuscript to emphasize that the failure of 3-IB-PP1 to initiate Erk/Akt signaling is not simply a failure to pass some threshold strength of membrane-proximal signaling. In addition, we have added a figure (Figure 4—figure supplement 2), which demonstrates that 3-IB-PP1 treatment actually suppresses Erk and Akt signaling through the CSF-1 receptor tyrosine kinase pathway, suggesting an active signaling block that prevents a full response to 3-IB-PP1 in unprimed cells.